# Arabidopsis *eIF4E1* protects the translational machinery during TuMV infection and restricts virus accumulation

Delyan Zafirov[1,2], Nathalie Giovinazzo[1], Cécile Lecampion[2], Ben Field[2], Julia Novion Ducassou[3], Yohann Couté[3], Karen S. Browning[4], Christophe Robaglia[2], Jean-Luc Gallois[1] *

1 GAFL, INRAE, Montfavet, France, 2 Aix-Marseille Univ, CEA, CNRS, BIAM, LGBP Team, Marseille, France, 3 Univ. Grenoble Alpes, INSERM, CEA, UA13 BGE, CNRS, CEA, Grenoble, France, 4 Department of Molecular Biosciences, The University of Texas at Austin, Austin, Texas, United States of America

* jean-luc.gallois@inrae.fr

**Data Availability Statement:** The mass spectrometry proteomics data have been deposited to the ProteomeXchange Consortium via the

## Abstract

Successful subversion of translation initiation factors eIF4E determines the infection success of potyviruses, the largest group of viruses affecting plants. In the natural variability of many plant species, resistance to potyvirus infection is provided by polymorphisms at *eIF4E* that renders them inadequate for virus hijacking but still functional in translation initiation. In crops where such natural resistance alleles are limited, the genetic inactivation of *eIF4E* has been proposed for the engineering of potyvirus resistance. However, recent findings indicate that knockout *eIF4E* alleles may be deleterious for plant health and could jeopardize resistance efficiency in comparison to functional resistance proteins. Here, we explored the cause of these adverse effects by studying the role of the *Arabidopsis eIF4E1*, whose inactivation was previously reported as conferring resistance to the potyvirus clover yellow vein virus (ClYVV) while also promoting susceptibility to another potyvirus turnip mosaic virus (TuMV). We report that *eIF4E1* is required to maintain global plant translation and to restrict TuMV accumulation during infection, and its absence is associated with a favoured virus multiplication over host translation. Furthermore, our findings show that, in the absence of eIF4E1, infection with TuMV results in the production of a truncated eIFiso4G1 protein. Finally, we demonstrate a role for eIFiso4G1 in TuMV accumulation and in supporting plant fitness during infection. These findings suggest that eIF4E1 counteracts the hijacking of the plant translational apparatus during TuMV infection and underscore the importance of preserving the functionality of translation initiation factors eIF4E when implementing potyvirus resistance strategies.

## Author summary

Plants are constantly under threat from viruses that can damage crops and reduce yield. Among these viruses, potyviruses are a major concern, and a small group of genes known as *eIF4E* are key factors in making a plant susceptible to them. To combat these viruses, it

PRIDE partner repository with the dataset identifier PXD036336.

**Funding:** Funding was provided by ANR-POTYMOVE (ANR-16-CE20-000803) to JLG. JND and YC acknowledge the support of Agence Nationale de la Recherche under projects ProFI (Proteomics French Infrastructure, ANR-10-INBS-08) and GRAL, a program from the Chemistry Biology Health (CBH) Graduate School of University Grenoble Alpes (ANR-17-EURE-0003). The funders had no role in study design, data collection and analysis, decision to publish, or preparation of the manuscript.

**Competing interests:** none.

is possible to either use naturally-selected variants of *eIF4E* that provide resistance, or to disable the gene altogether. However, new research has shown that inactivating *eIF4E* genes may have unintended consequences for the plant's development while compromise resistance to other potyviruses. To investigate this further, we focus in this work on the role of the *Arabidopsis eIF4E1* whose inactivation confers resistance to one potyvirus, clover yellow vein virus (ClYVV). We looked why this same mutation at *eIF4E1* makes the plants more susceptible to another potyvirus, turnip mosaic virus (TuMV). Our study reveals that eIF4E1 acts in safeguarding the plant translational machinery during TuMV infection. By preventing the degradation of the translation initiation protein eIFiso4G1, eIF4E1 enables the plant to maintain its normal translation activity and ultimately prevents the accumulation of virus proteins. Our findings provide valuable insights into how potyviruses hijack the plant's translation process, and emphasize the need of preserving the functionality of translation initiation factors when developing potyvirus resistances.

## Introduction

Translation, the decoding of a messenger RNA (mRNA) into a protein product, is a fundamental process in any cell that comes at a high energetic price and is therefore subjected to tight regulatory mechanisms [1,2]. At the same time, translation is parasitized by viruses, obligatory intracellular infectious agents whose successful infection and dissemination is completely dependent on the translational apparatus of the host cell [3–5]. In eukaryotes, the initiation of the translation process is assured by at least 16 different translation initiation factors (eIFs), among which the eIF4F complex plays a key role in mediating the loading of mRNA on ribosomes [6]. Conserved among plants, fungi and metazoans, this heterotrimeric complex is composed of the eIF4E, eIF4G and eIF4A proteins: eIF4E is a cap-binding protein that interacts with the methyl-7-guanosine (5' cap) structure of mRNA, the eIF4A helicase unwinds secondary structures at the 5' end of mRNA and the eIF4G subunit serves as a scaffold that engages in protein-protein interactions culminating in the recruitment of the small 40S ribosome to the mRNA [1,7] (Fig 1a).

   The eIF4F complex is subject to diverse regulatory mechanisms that control the overall translational landscape in the cell and is also a target hijacked by viruses [8]. In animals and fungi, 4E-binding proteins (4E-BP), small proteins regulated by the mechanistic target of rapamycin 1 (mTORC1) kinase, influence the overall translation activity by modulating the formation of the eIF4F complex [9]. The encephalomyocarditis virus (EMCV), responsible for myocarditis and encephalitis in mammals, induces the accumulation of hypophosphorylated 4E-BPs which disrupt the assembly of a functional eIF4F complex for translation initiation [10,11]. The MAPK-interacting kinases Mnk1 and Mnk2 induce the phosphorylation of mammalian eIF4E at Serine 209, leading to translational reprogramming and the selective translation of specific mRNA subsets [12,13]. Animal viruses alter the phosphorylation state of eIF4E during infection: while the calicivirus murine norovirus 1 (MNV1) and the herpes simplex virus 1 (HSV1) trigger eIF4E phosphorylation [14,15], others such as adenovirus and the vesicular stomatitis virus (VSV) are known to dephosphorylate eIF4E [16,17]. An aggressive strategy of eIF4F complex subversion is well documented for poliovirus (PV), the picornavirus responsible for paralytic poliomyelitis in humans, which employs virus-encoded proteases to cleave mammalian eIF4G proteins during infection [18–20]. By co-opting the eIF4F complex, viruses alter the translational landscape in the cell and in some cases may completely shut off host translation–the production of antiviral proteins is thus inhibited and the cellular activity

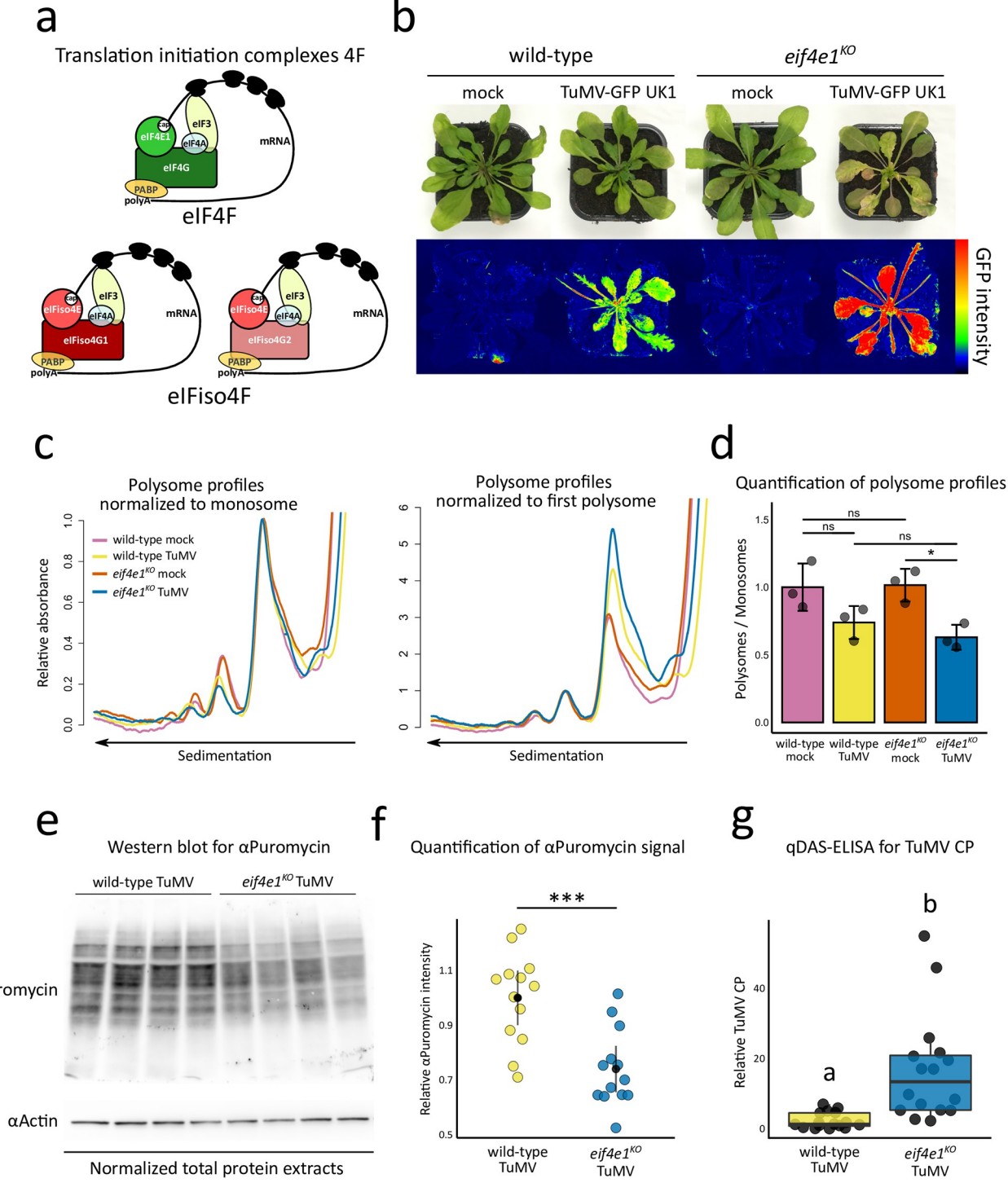

**Fig 1. In the absence of *eIF4E1*, global translation activity decreases while virus accumulation increases. (a)** Graphical representation of the translation initiation complexes 4F in *Arabidopsis thaliana*. **(b)** Phenotypic comparison of representative wild-type and *eif4e1^KO^* plants in the absence (mock) or presence of TuMV-GFP UK1 infection 14 dpi. Photographs were taken under natural light conditions (upper panel) and under wavelengths specific for green fluorescent protein (GFP) excitation (lower panel) by using GFP fluorometric imaging. **(c)** Comparison of monosome-normalized (left panel) and first polysome-normalized (right panel) polysome profiles 14 dpi. n = 3 biological replicates. **(d)** Quantifications of polysome to monosome ratio from raw absorbance profiles. Error bars indicate standard deviation. n = 3 biological replicates **(e)** Comparison of puromycin incorporation 14 dpi. Western blot using anti-puromycin antibodies was performed on 20 ug of total protein extracts and equal protein levels were checked using anti-Actin antibodies. **(f)** Quantification of puromycin signal intensity. Graphs show the 95% confidence intervals of the means obtained

from three independent experimental repeats. n = 13 biological replicates. (g) Accumulation analysis of TuMV by a quantitative DAS-ELISA at 14 dpi. n = 16 biological replicates. "ns" non-statistically significant difference, "*" $P < 0.05$, "**" $P < 0.01$, "***" $P < 0.001$. Different letters depict significantly different groups identified by Kruskal–Wallis at $P < 0.05$.

is allocated to virus translation which proceeds via alternative translation mechanisms [21]. As a critical virus target, controlling the availability or function of eIF4F has been pointed out as a potential therapeutic strategy to limit viral disease in humans [22–27].

In plants, comparatively little is known about regulatory mechanisms at the level of eIF4F and how they impact translation [28–30]. However, plant viruses, especially members of the *Potyviridae* family clearly require eIF4F for their replication, as illustrated by the existence of numerous alleles of the complex providing virus resistance in model and cultivated crops [4]. In addition to the conserved eIF4F complex, a closely related eIFiso4F complex, composed of cap-binding eIFiso4E and scaffolding eIFiso4G proteins is also present in flowering plants [31,32]. Throughout the text we refer to the eIF4F and eIFiso4F complexes as 4F. We use "4E" to denominate eIF4E/eIFiso4E subunits and "4G" for the eIF4G and eIFiso4G subunits. The two 4F complexes exhibit a significant degree of functional redundancy as single *4E/4G* mutations moderately affect global plant development (Fig 1a and 1b). The relationship between the plant 4F complex and potyviruses was first discovered over 25 years ago when it was found that a viral genome-linked protein (VPg) covalently attached to the 5' end of the turnip mosaic virus (TuMV) genome can bind the *Arabidopsis* translation initiation factor eIFiso4E [33]. The VPg-eIFiso4E interaction is critical for infection success as the inactivation of *eIFiso4E* in *Arabidopsis* confers complete resistance to TuMV [34,35]. Since then, genetic and biochemical studies have found that in numerous plant species, naturally occurring polymorphisms at *eIF4E* alleles translate to amino acid substitutions that provide virus resistance due to disrupted interactions with VPg or other viral factors [4,36]. Not only do these 4E resistance proteins successfully evade virus recruitment, but in most cases they also retain complete functionality in the translation initiation process [37–39], (for review [32]). In recent years, the advent of genome editing techniques such as CRISPR-Cas9 has found a successful application in the introduction of virus resistance into crops devoid of resistance mechanisms through knocking out *4E* or *4G* genes [40–44] or by modifying their function in order to escape virus hijacking [45–47].

Direct comparisons between functional and knockout *eIF4E* resistance alleles have indicated that proteins retaining their translation activity may be a source of more efficient potyvirus resistance [32]. For instance, in tomato, the functional *Sh-eIF4E1$^{PI24}$-pot1* allele has been found to provide a broader potyvirus resistance spectrum in comparison to an *eIF4E1* knockout allele [39]. Another example comes from the pepper (*Capsicum annuum*)–PVY pathosystem, where the *pvr6* allele coding for a non-functional eIFiso4E protein was suggested to associate with lower resistance durability compared to the functional *pvr6$^+$* allele [48]. These observations coming from two major crop species suggest that natural or artificial selection did not favour knockout alleles over proteins retaining biological activity. In support of this, we have recently found that in *Arabidopsis* the inactivation of *eIF4E1* to generate resistance to clover yellow vein virus (ClYVV) is responsible for a remarkable increase in susceptibility to the potyvirus TuMV which requires eIFiso4E for accumulation [49]. This hypersusceptibility is characterized by the development of intense disease symptoms, including growth arrest and increased senescence, as well as a strong overaccumulation of TuMV proteins throughout the plant (Fig 1b). In contrast, developing a functional *eIF4E1* resistance allele by CRISPR-Cas9 base editing allowed to avoid the TuMV hypersusceptibility phenotype while preserving ClYVV resistance [46,49].

Here we studied the mechanism responsible for the aforementioned induction of TuMV hypersusceptibility in the absence *eIF4E1* in *Arabidopsis*. By analysing the translation activity, integrity of translation initiation proteins and virus load in wild-type and *eif4e1$^{KO}$* healthy or TuMV-infected plants, we uncover a role for eIF4E1 in preserving the global host translational landscape, safeguarding the translation machinery and decreasing virus load during infection. We further show that eIFiso4G1 is a critical protein that is required for TuMV accumulation and acts in concert with eIF4E1 to sustain plant health on TuMV infection. Our findings identify a crosstalk between plant 4F complexes which controls both host translation and potyvirus susceptibility.

## Results

### The absence of eIF4E1 leads to an increase in TuMV protein synthesis while overall plant protein synthesis is reduced

We hypothesized that the exacerbated symptoms that develop in *eif4e1$^{KO}$* plants infected with TuMV may be related to alterations in host translation activity. To investigate this, we set out to explore the impact of TuMV on host translation in wild-type and *eif4e1$^{KO}$* plants (Fig 1).

We first examined translation activity using polysome profiling, a method that allows analysis of the global level of translation by the generation of absorbance profiles that reflect the distribution of ribosomes on transcripts [50]. The obtained absorbance profiles can be normalized to either the monosome peak–which allows comparison of the levels of polysomes (actively translated transcripts), or to the first polysome peak–to compare the abundance of monosomes (non-actively translated transcripts) [51]. To evaluate the consequence of TuMV infection, we first checked whether wild-type and *eif4e1$^{KO}$* plants display any changes in global translation activity in the absence of virus infection (Fig 1c and 1d and S1 Fig). We did not observe any major differences in levels of polysomes or monosomes between mock-inoculated wild-type and *eif4e1$^{KO}$* plants (Fig 1c, **magenta and vermilion profiles**, S1a Fig). In agreement, quantification of the area under polysome and monosome peaks showed that there were no significant changes between these plants (Tukey's HSD test, $P = 0.9990$) (Fig 1d). These results suggest that in the absence of virus infection, the inactivation of *eIF4E1* does not modify the global translation of the plant. We extended our analysis on plants infected with TuMV at 14 days post-inoculation (dpi). In wild-type plants, the average polysome/monosome ratio was lower by 26% during TuMV infection, although the difference with the mock-inoculated control was not significant (Tukey's HSD test, $P = 0.1450$) (S1b and S1c Fig). In contrast, in the absence of *eIF4E1*, TuMV infection induced a significant 38% decrease in the polysome/monosome ratio compared to mock-inoculated plants (Tukey's HSD test, $P = 0.0282$) (Fig 1d). This was characterized by lower polysome levels and increased monosome accumulation, signatures of a translational impediment [52–54] (Fig 1c, **blue profiles**, S1d Fig). Overall, more experiments would be needed to confirm whether TuMV causes a decrease in translation activity in wild-type plants, however this effect is very clear in the absence of *eIF4E1*.

Next, we used an independent approach to determine the effect of *eIF4E1* loss on translation activity during TuMV infection. To do this, we measured the levels of global translation in plants by treating them with puromycin, an aminoacyl tRNA analog that is incorporated into nascent polypeptide chains during translation and can serve as a proxy for the global protein synthesis rate [50,55,56] (Fig 1e and 1f, S2 and S3 Figs). The level of puromycin incorporation into nascent proteins in TuMV-inoculated *eif4e1$^{KO}$* versus mock-inoculated controls was reduced compared to wild-type plants, which is consistent with the above-mentioned polysome profiling results (S2 Fig, **compare left and right panels**). To firmly validate the stronger translational defect imposed by TuMV in the absence of *eIF4E1*, we then directly

quantified the ratio of puromycin incorporated into wild-type relative to *eif4e1*$^{KO}$ plants in the absence or presence of TuMV in at least two independent experimental replicates (Fig 1e and 1f, S3 Fig). In agreement with our previous results, TuMV infection resulted in a significant (two-way ANOVA, Tukey, *P* = 0.0003) decrease of approximatively 30% in the global protein synthesis rate in *eif4e1*$^{KO}$ plants compared to wild-type plants (Fig 1f and S3c Fig), while no significant changes were detected between mock-inoculated wild-type and *eif4e1*$^{KO}$ plants (two-way ANOVA, Tukey, *P* = 0.314) (S3a and S3b Fig). Thus, our analyses of host translation by two independent approaches demonstrate that TuMV imposes a greater reduction in the global translation activity in the *eif4e1*$^{KO}$ mutant relative to wild-type plants.

We found that the enhanced TuMV susceptibility of *eif4e1*$^{KO}$ plants was also associated with higher virus accumulation, as attested by imaging of TuMV-GFP infected plants (Fig 1b). To confirm this, we performed a quantitative DAS-ELISA test to precisely determine the concentration of TuMV coat protein (CP) in wild-type and *eif4e1*$^{KO}$ infected plants 14 dpi. This analysis showed that the TuMV load was about 7 times higher in average in *eif4e1*$^{KO}$ plants compared to wild-type plants (Fig 1g).

These data suggest that, in the absence of *eIF4E1*, while global translation is reduced, virus translation is simultaneously increased.

## TuMV infection reduces the abundance of specific components of the plant 4F translation initiation complexes in *eif4e1*$^{KO}$

We next sought to understand the cause for the decreased host translation activity observed in the *eif4e1*$^{KO}$ mutant during TuMV infection. The initiation of translation is considered an important regulatory step whose efficiency influences the entire translation process [57]. We therefore focused on the components of the translation initiation complexes eIF4F and eIFiso4F, whose importance in the potyvirus infection cycle is well documented (Fig 1a) [4,7]. To analyse the implication of 4E and 4G proteins in the translation initiation process, we compared their relative abundance by mass spectrometry (MS)-based quantitative proteomics following a mRNA cap analogue pull-down assay on protein extracts obtained from wild-type and *eif4e1*$^{KO}$ mutant plants infected or not with TuMV for 21 days (Fig 2, S1 Table, **Proteins**).

As an experimental control, we checked that eIF4E1 was not detected in *eif4e1*$^{KO}$ plants whereas it was identified and quantified in wild-type plants (Fig 2a and 2b and S1 Table, **Proteins**). In wild-type plants, the abundance of 4E (eIF4E1 and eIFiso4E) and 4G (eIF4G, eIFiso4G1 and eIFiso4G2) proteins was not significantly different between mock and TuMV-infected plants (Fig 2c). In *eif4e1*$^{KO}$ plants, we observed important changes in the abundance of several 4F components. First, eIF4G was mostly under the detection limit in *eif4e1*$^{KO}$ samples in the absence or presence of TuMV (Fig 2a and 2b, detected once with low abundance out of six samples, S1 Table, **Proteins**). This suggests that, in *eif4e1*$^{KO}$, eIF4G is not present in cap binding complexes and that only eIFiso4F is functionally available to carry out the translation initiation process. Second, during TuMV infection, the abundance of the translation initiation proteins eIFiso4G1 and eIFiso4G2 was significantly decreased by a factor of 19.5 and 8.5, respectively (ANOVA Benjamini-Hochberg q-value < 0.05, and *limma* p-values < 0.0001) (Fig 2d). These observations suggest that only a small portion of eIFiso4F complexes is available for translation initiation in TuMV-infected *eif4e1*$^{KO}$ plants.

Collectively, these results suggest that the presence of eIF4E1 counteracts the targeting of specific translation initiation proteins following TuMV infection. The inhibition of host translation in TuMV-infected *eif4e1*$^{KO}$ plants thus appears to be associated with a direct impairment of the translation initiation apparatus resulting from two factors: the unavailability of eIF4F complexes required for protein synthesis during infection and a considerable

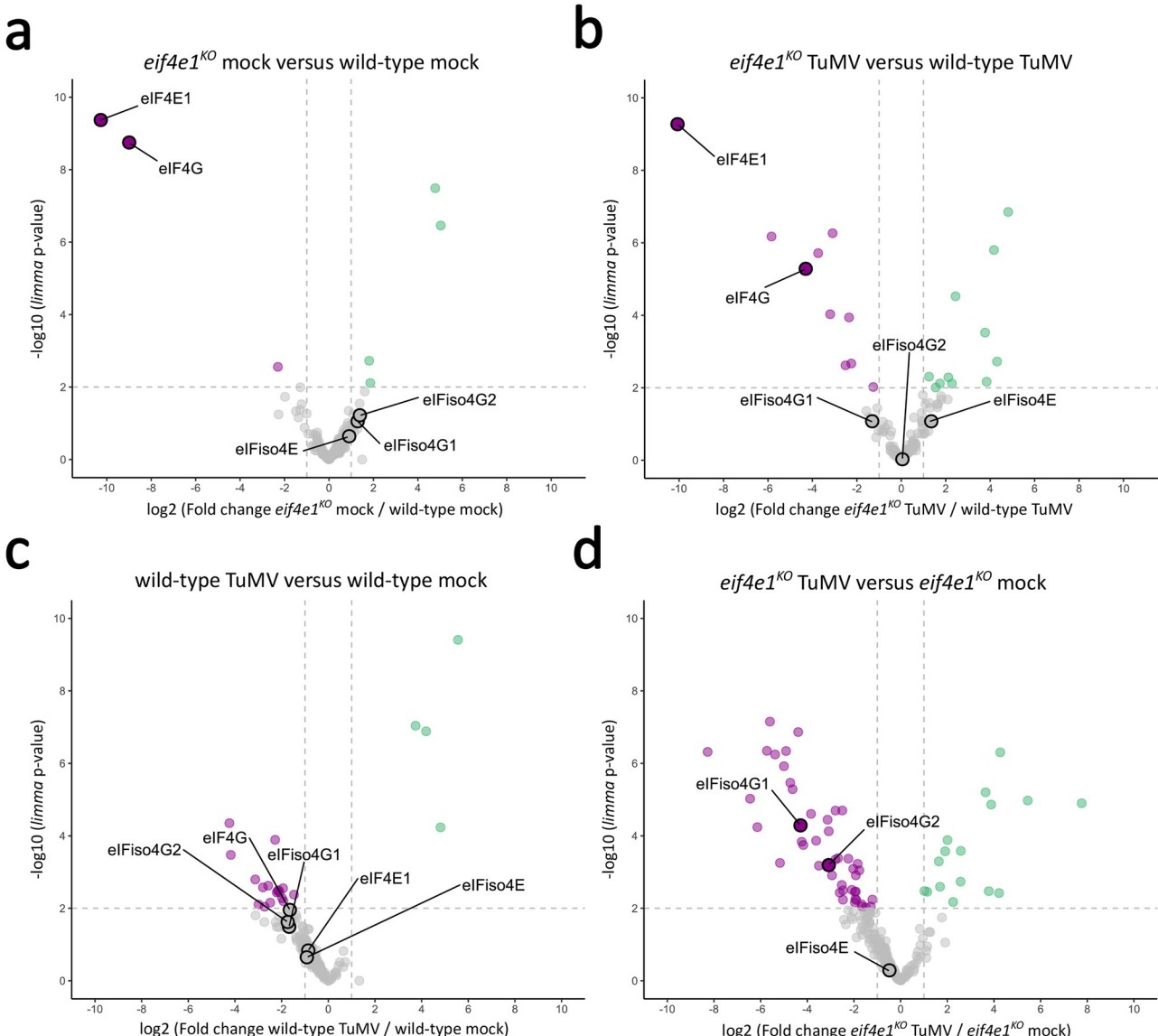

**Fig 2. TuMV modifies the translational initiation machinery in the absence of *eIF4E1*.** Volcano plots displaying the differential abundance of proteins purified using m7GTP pull-down and analysed by MS-based label-free quantitative proteomics in mock-inoculated *eif4e1^{KO}* versus wild-type (**a**) TuMV-inoculated *eif4e1^{KO}* versus wild-type (**b**), mock versus TuMV-inoculated wild-type (**c**) and mock versus TuMV-inoculated *eif4e1^{KO}* plants (**d**). The volcano plots represent the -log10 (limma p-value) on y axis plotted against the log2 (Fold Change) on x axis for each quantified proteins. Green and magenta dots represent significantly enriched proteins (log2 (Fold Change) ≥ 1 and -log10(p-value) ≥ 2) whereas grey dots represent those with insignificant changes. Only proteins found enriched in m7GTP pull-down samples compared to negative control samples are represented. n = 3 biological replicates.

reduction in the levels of eIFiso4G1 and eIFiso4G2 proteins present in the translation initiation complexes.

## eIF4E1 is required for the integrity of eIFiso4G1 during TuMV infection

The strong decrease in the abundance of eIFiso4G1 in the translation initiation complexes of TuMV-infected *eif4e1^{KO}* plants prompted us to further analyse the possible causes for it.

Therefore, we examined the distribution of eIFiso4G1-matching peptides identified by MS-based proteomics to the eIFiso4G1 protein sequence in wild-type and *eif4e1^KO* plants in the absence and presence of TuMV infection (Fig 3a–3d). In both wild-type and *eif4e1^KO* mock-inoculated plants, peptides covering the entire protein sequence of eIFiso4G1, including the eIFiso4E binding, HEAT1/MIFG, and HEAT2/MA3 domains, were identified (Fig 3a and 3b).

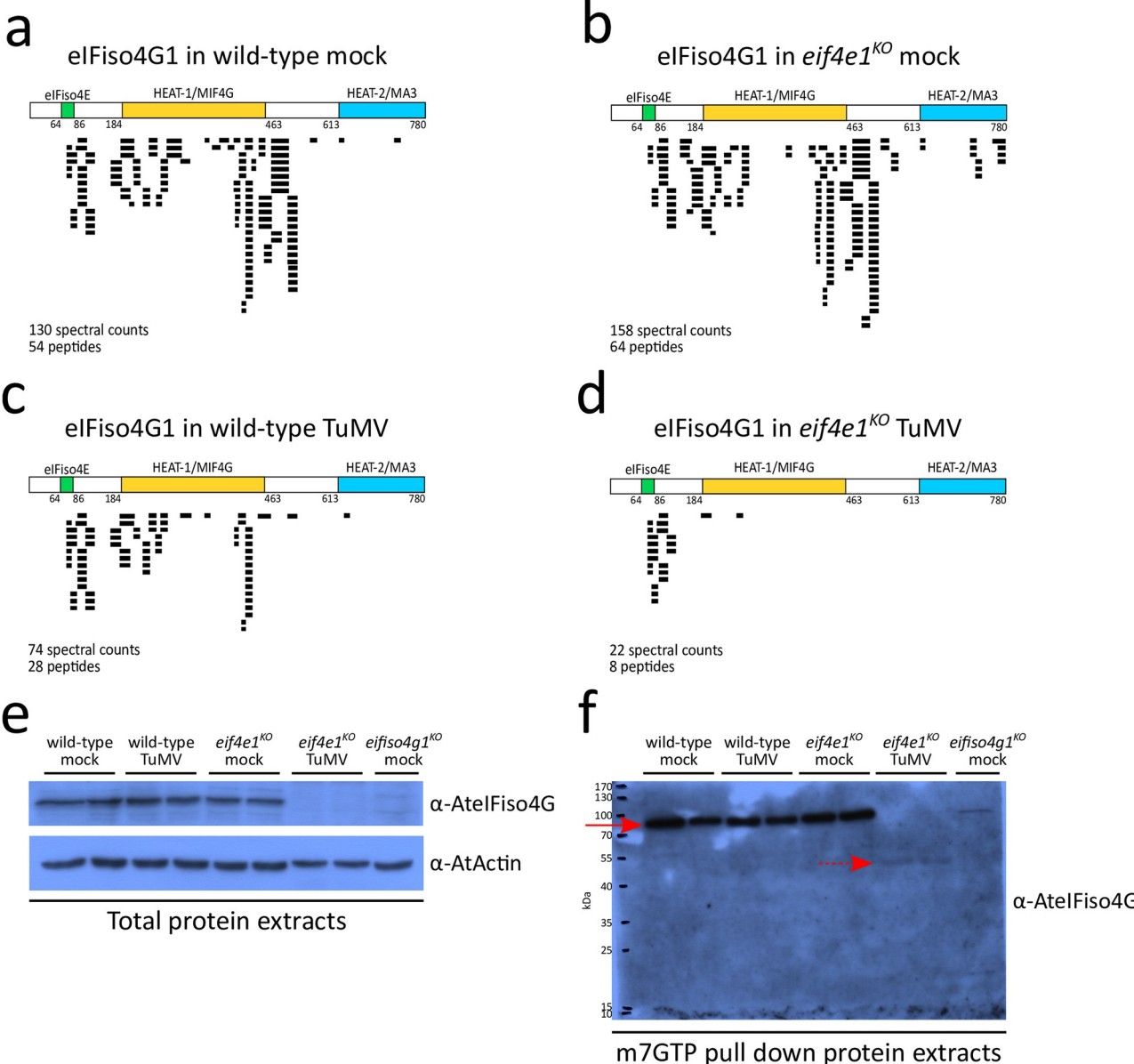

**Fig 3. In the absence of *eIF4E1*, TuMV infection induces the production of a truncated version of eIFiso4G1. (a-d)** Graphic representations of the coverage of eIFiso4G1 protein sequence obtained by MS-based proteomic analyses. The upper panel proposes a schematic representation of eIF4G1 sequence with the positions of the eIFiso4E-binding, HEAT1/MIF4G and HEAT2/MA3 domains. Below, each horizontal black bar corresponds to an identified peptide spectral count matching eIFiso4G1 sequence and is placed according to the position of the identified peptide in the protein sequence. The total number of peptides and spectral counts identified in each experimental condition are given. n = 3 biological replicates. **(e-f)** Western blot analyses of eIFiso4G1 in total protein extracts **(e)** or m7GTP pull down protein extracts **(f)** 14 dpi. Red arrowheads indicate the position of the full-length, 86 kDa, eIFiso4G1 protein. Dashed red arrowheads indicate the position of a 50 kDa protein product recognized by anti-eIFiso4G antibodies. Equal loading was checked by western blot using anti-actin antibodies.

In contrast, TuMV infection caused a reduction in the number of identified peptides along the eIFiso4G1 sequence, consistent with the decrease in protein abundance. In the infected wild type, peptides for eIFiso4G1 were distributed all along protein sequence (Fig 3c). Remarkably, in TuMV infected *eif4e1*$^{KO}$ plants, we only detected peptides matching the N-terminal region of eIFiso4G1, i.e. spanning the eIFiso4E binding domain and a portion of the HEAT1/MIFG domain (Fig 3d).

In view of these results, we reasoned that the strong reduction in the abundance of eIFiso4G1 protein in TuMV-infected *eif4e1*$^{KO}$ plants could be due to cleavage or degradation of eIFiso4G1 resulting in the loss of the C-terminal part of the protein. To test this, we monitored the accumulation and integrity of eIFiso4G1 in total and m7GTP pull-down protein extracts from mock and TuMV-infected plants (Fig 3e and 3f). In total protein extracts, the full-length eIFiso4G1 protein accumulated at similar levels in both mock wild-type and *eif4e1*$^{KO}$ plants, as well as in TuMV-inoculated wild-type plants. However, the full-length eIFiso4G1 was barely detectable in TuMV-inoculated *eif4e1*$^{KO}$ plants (Fig 3e and S4a Fig). In m7GTP pull-down protein extracts, similar results were obtained for the full-length eIFiso4G1 protein, migrating at ~86 kDa (Fig 3f and S4b Fig, **red arrows**). Strikingly, in addition to the loss of the full-length eIFiso4G1 a band at approximately 50 kDa was detected in the TuMV-infected *eif4e1*$^{KO}$ samples (Fig 3f and S4b Fig, **dashed red arrows**). This eIFiso4G1-antibody immunoreactive band may correspond to a truncated eIFiso4G1 protein, as suggested by the peptide distribution profile (Fig 3d). To test whether this is the case, we identified proteins by MS in the 40 and 55 kDa region in m7GTP pull-down extracts obtained from both mock and TuMV-inoculated *eif4e1*$^{KO}$ plants (S5a Fig). We detected peptides covering the eIFiso4E-binding and HEAT1/MIFG domains of eIFiso4G1 in *eif4e1*$^{KO}$ TuMV-inoculated samples, confirming the existence of a C-terminal truncated eIFiso4G1 form at this position (S5b Fig). However, although our above western blot analyses did not reveal it, we still detected a few eIFiso4G1-matching peptides in *eif4e1*$^{KO}$ mock samples (S5c Fig). This could indicate that truncated versions of eIFiso4G1 protein exist at low accumulation levels in the absence of virus infection. Finally, no amino acid consensus sequences that could be recognized and cleaved by one of the three TuMV-encoded proteases (NIa-Pro, P1 and HC-Pro) were found on the eIFiso4G1 protein sequence, even when the search parameters were adjusted to include non-canonical consensus cleavage sequences (S6 Fig), suggesting that the viral proteases were not responsible for the degradation event.

In summary, our data demonstrate that in the absence of *eIF4E1*, TuMV strongly decreases the accumulation of the full-length, intact, eIFiso4G1 protein and exacerbates the production of a truncated eIFiso4G1 protein containing only the N-terminal eIFiso4E binding and HEAT1/MIFG domains.

### *eIFiso4G1* inactivation during TuMV infection is associated with increased growth inhibition and reduced virus accumulation

Our proteomic and immunoblot data point to a TuMV-exacerbated degradation of eIFiso4G1 that occurs specifically in plants knocked out for *eIF4E1*. We therefore wondered whether this eIFiso4G1 degradation is responsible for the strong phenotype and increased TuMV accumulation in these plants. To test this, we investigated the role of *eIFiso4G1* in the *eIF4E1* knock-out-mediated TuMV oversusceptibility by analysing the susceptibility phenotypes of single and double *Arabidopsis* mutants for these two genes (Fig 4, S7 and S8 Figs).

We first compared the virus-induced disease symptoms in wild-type, *eifiso4g1*$^{KO}$, *eif4e1*$^{KO}$, *cum1* (an ethyl methanesulfonate (EMS) mutagenesis generated *eIF4E1* null allele, Experimental procedures) and *cum1 eifiso4g1*$^{KO}$ plants 14 dpi with TuMV. Infected wild-type plants

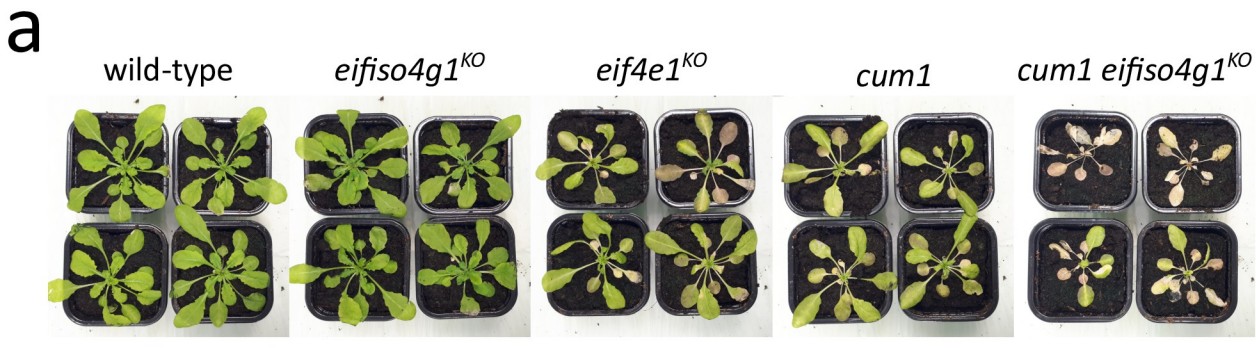

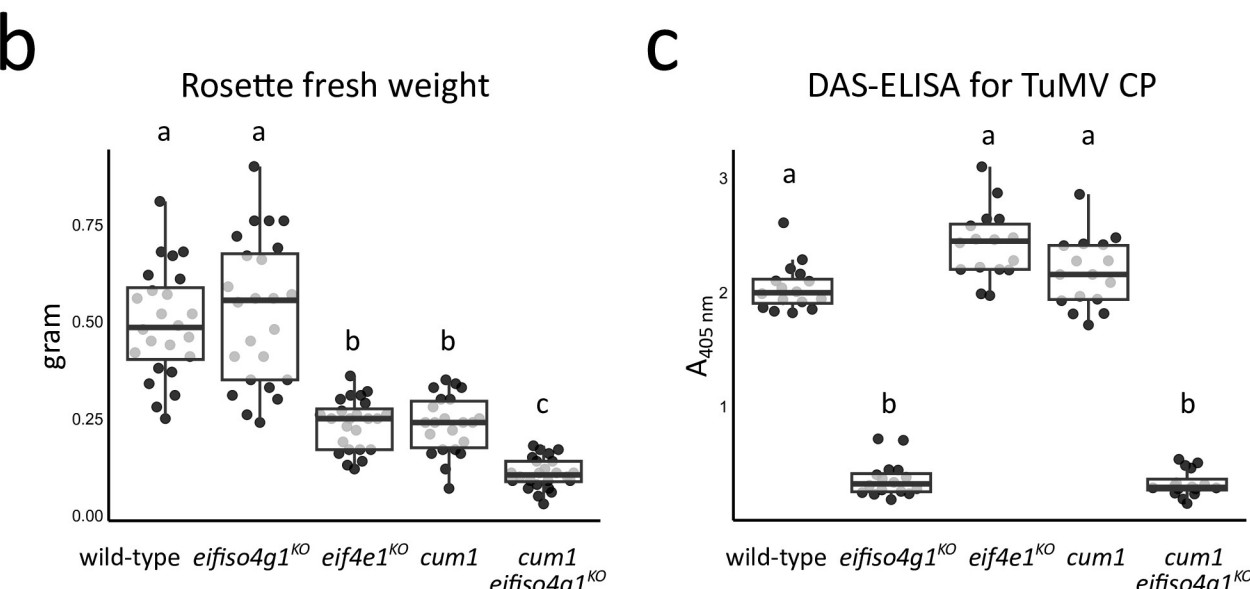

**Fig 4. The combined absence of both *eIF4E1* and *eIFiso4G1* enhances TuMV disease symptoms but impairs TuMV accumulation. (a)** Phenotypic comparison of TuMV-inoculated representative plants of each genotype 14 dpi. **(b)** Rosette fresh weight analysis of TuMV-inoculated plants 14 dpi. n = at least 22 biological replicates. **(c)** Accumulation analysis of TuMV by DAS-ELISA 14 dpi. n = 16 biological replicates. Different letters depict significantly different groups identified by Kruskal–Wallis statistical tests at $P < 0.05$.

displayed the common TuMV disease symptoms, including leaf chlorosis and curling, and stunted growth (Fig 4a). In *eifiso4g1*KO plants, the phenotypic responses appeared to be similar to wild-type, as attested by visual inspection of disease symptoms and rosette fresh weight analysis (Fig 4a and 4b, S7a and S7b Fig). However, as we previously reported [49], the disease symptoms and the loss of rosette fresh weight were strongly enhanced in TuMV-infected *eif4e1*KO plants (Fig 4a and 4b, S7a and S7b Fig). We also observed an identical marked phenotypic response in the *cum1* mutant, an independent null *eIF4E1* allele (Fig 4a and 4b, S7a and S7b Fig) [58]. This further demonstrates that the strongest symptoms are caused by the lack of eIF4E1. We next looked at the effect of combining loss-of-function mutations in both *eIF4E1* and *eIFiso4G1*. Strikingly, the intensity of the disease symptoms in the *cum1 eifiso4g1*KO line was further increased relative to the lines disrupted for *eIF4E1* or *eIFiso4G1* alone, as evidenced by the considerable growth reduction at young leaves and pronounced tissue senescence that

spread throughout all aerial organs, resulting in plant death in approximatively half of the *cum1 eifiso4g1^KO* TuMV-inoculated plants (Fig 4a, S7a Fig). Accordingly, *cum1 eifiso4g1^KO* displayed a significant decrease of approximatively 50% in rosette fresh weight compared to lines disrupted for *eIF4E1* (Fig 4b, S7b Fig). Thus, the absence of *eIFiso4G1* on its own does not have a role in the development of TuMV disease symptoms, but strongly increases their severity when combined with *eIF4E1* loss. To examine overall plant health in the absence of virus infection, we recorded the phenotypes of the same set of mock-inoculated plants 14 dpi (S8 Fig). As previously noted, the single mutants *eifiso4g1^KO*, *eif4e1^KO* and *cum1* presented a slightly smaller rosette stature and reduced growth in comparison to wild-type [59–61]. The double mutant *cum1 eifiso4g1^KO* displayed an approximatively 2 times decrease in its rosette fresh weight relative to wild-type. However, there were no significant differences in the weight of the double mutant compared to the single mutants affected for either *eIF4E1* or *eIFiso4G1* genes alone. Moreover, mock-inoculated *cum1 eifiso4g1^KO* did not exhibit the striking developmental deformations and tissue senescence as seen on TuMV infection (S8a Fig). Hence, the strong TuMV susceptibility phenotype of *cum1 eifiso4g1^KO* cannot be solely attributed to the combined absence of eIF4E1 and eIFiso4G1 proteins but appears to be specific to virus infection.

To assess the role of eIFiso4G1 in the accumulation of TuMV, we then compared the TuMV protein load in the different *Arabidopsis* genotypes at 14 dpi using DAS-ELISA tests for TuMV CP quantification (Fig 4c, S7c Fig). Consistent with our previous results [49], TuMV accumulation increased in both lines disrupted for *eIF4E1* -namely *eif4e1^KO* and *cum1*—compared to wild-type plants. Surprisingly, the TuMV CP titers were significantly decreased by about 10 times in *eifiso4g1^KO* plants relative to wild-type, suggesting that a functional eIFiso4G1 protein is necessary for the accumulation of TuMV. TuMV accumulation was strongly decreased in *cum1 eifiso4g1^KO* plants, similar to what is observed in plants disrupted for *eIFiso4G1* alone. These results indicate that eIFiso4G1 is an important factor for TuMV accumulation in the presence or absence of *eIF4E1*.

Thus, the removal of *eIFiso4G1* in *eif4e1^KO* further exacerbates the growth defects and the induction of senescence observed upon TuMV infection, but is at the same time associated with reduced TuMV load. These data suggest that eIFiso4G1 may be directly involved in the hypersusceptibility of the *eif4e1^KO* mutant towards TuMV by preventing the induction of strong disease symptoms and supporting TuMV multiplication.

## Discussion

As genes that confer resistance or susceptibility to pathogens may also have roles that are not directly related to the pathogen infection success, it is important to consider the potential pleiotropic effects when designing genetic resistances [62]. Translation initiation factors exhibit a dual role in protein synthesis and virus susceptibility and thus offer a good model to understand how these genes should be manipulated to ensure effective resistance without side effects. Here, we undertook the examination of a phenomenon observed in *Arabidopsis* in which the inactivation of the *eIF4E1* gene does not only provide resistance to some plant viruses such as the potyvirus ClYVV [63], the cucumovirus cucumber mosaic virus (CMV) [58] and the poleroviruses beet mild yellowing virus (BMYV) and beet western yellows virus-USA (BWYV-USA) [64], but also triggers increased susceptibility to the potyvirus TuMV. Behind this hypersusceptibility, we unveil a host hijacking scheme that operates at the translational level. During TuMV infection, eIFiso4G1, the scaffolding protein for the eIFiso4F translation initiation complex, is degraded in a process associated with decreased global plant translation but increased TuMV multiplication. The cap-binding protein eIF4E1 of the eIF4F

complex is central to counter this virus usurpation strategy as its presence alleviates the translational repression and restricts TuMV accumulation. This previously undocumented potyvirus strategy for subverting the plant translational apparatus emphasizes the importance of maintaining a balance between eIF4F and eIFiso4F components in order to avoid increased potyvirus susceptibility.

Given the duality of translation initiation factors in mRNA translation and potyvirus susceptibility, a long-lasting question has been how potyvirus-mediated subversion of 4E and 4G proteins impacts the host translation. Early studies argued that the potyvirus protein VPg could negatively regulate the function of plant translation initiation factors [65–68]. More recently, this suggestion was re-evaluated through detailed biochemical and structural studies that revealed a competitive binding of the VPg protein of PVY to the cap-binding pocket of the human eIF4E protein [69]. As the presence of VPg decreased the translation of capped mRNA in wheat germ extracts and human osteosarcoma cancer cells, VPg was suggested to sequester eIF4E away from host mRNA translation [69]. These effects observed in heterologous *in vitro* and *in vivo* systems are also mirrored in plant cells: transient expression of potato virus A (PVA) VPg in *Nicotiana benthamiana* leaves downregulated the translation of co-expressed luciferase reporter mRNA constructs [70]. Interestingly, however, it has not been observed that host translation is globally affected during natural potyvirus infection *in planta*. Polysome profiling analyses performed on TuMV-infected *Arabidopsis* and PVA-infected *N. benthamiana* plants have revealed only a minor attenuation of the host translation that did not alter the overall translatome, but rather remodelled the translation of specific mRNA sets [71,72]. Here, our findings which are supported by both polysome profiling and puromycin incorporation analyses, align with the above-mentioned studies. Moreover, our data suggest that eIF4E1 is critical in maintaining this balance of translation in *Arabidopsis*, as its absence was associated with compromised host protein synthesis following TuMV infection.

Perhaps more intriguingly, the repressed global translation in TuMV-infected *eif4e1*$^{KO}$ plants was associated with the production of a truncated form of eIFiso4G1 protein. To our knowledge, this is the first documentation of a potyvirus-induced modification of a translation initiation factor in plants. In animals, degradation of eIF4G proteins is known to occur upon infection by several picornaviruses, including PV, human rhinovirus (HRV), coxsackie B virus (CVB3) and foot-and-mouth-disease virus (FMDV) [73–75]. These viruses employ 2A protease proteins to cleave the animal eIF4G paralogs into two fragments–an N-terminal fragment containing the eIF4E-binding domain and a C-terminal fragment comprising the domains necessary for 40S ribosome recruitment [73]. Inspired by the similarity with these models, we searched the eIFiso4G1 protein for the presence of specific amino acid sequences that could be recognized by a TuMV-encoded protease. As we did not find any potential potyvirus protease cleavage site on eIFiso4G1 and we detected low eIFiso4G1 degradation even in the absence of TuMV infection, we propose that an endogenous degradation mechanism stimulated by TuMV is more likely to underlie the production of the truncated eIFiso4G1 protein. Indeed, a recent study combining computational and functional approaches did not uncover an obvious cleavage site on the *Arabidopsis* eIFiso4G1 that could be targeted for proteolysis by the TuMV protease NIa-Pro [76]. While we propose that the proteolytic processing of eIFiso4G1 is the probable event causing the production of a truncated protein form, it is also possible that alternative mechanisms operating at the transcriptional level (such as changes in transcription or splicing patterns) or even unconventional translation initiation (like translation initiation at non-canonical sites) could also give rise to shorter eIFiso4G1 protein variants.

A key question that our work raises is what might be the biological significance of the accumulation of truncated forms of eIFiso4G1, albeit at much lower amounts than full-length protein, for host translation and TuMV accumulation. The eIFiso4G1 protein recovered by

m7GTP pull-down assays on TuMV-infected *eif4e1*<sup>KO</sup> plants contains the N-terminal region that comprises the eIFiso4E binding site and the HEAT1/MIF4G domain but lacks the C-terminally situated HEAT2/MA3 domain. In the wheat (*Triticum aestivum*) eIFiso4G1 paralog, the eIFiso4E binding site recruits the eIFiso4E protein, while the HEAT1/MIF4G and HEAT2/MA3 domains are required for eIF4A, eIF4B, PABP and eIF3 loading [77,78], (for review [79]). The deletion of the HEAT2/MA3 in wheat eIFiso4G1 was found to disrupt interactions with the eIFiso4G1-binding partners PABP, eIF4A and eIF4B that play a critical role in circularisation and unwinding of eIFiso4F-bound mRNAs [77]. Moreover, wheat eIFiso4G1 lacking the entire HEAT-2 domain and C-terminal regions of the HEAT-1 domain was impaired in its translation activity *in vitro* [78]. Therefore, it is plausible that a diminished efficiency of the truncated *Arabidopsis* eIFiso4G1 protein to support protein synthesis, along with the complete absence of eIF4F complexes for translation initiation, are responsible for the decrease in host translation observed in the TuMV-infected *eif4e1*<sup>KO</sup> plants.

Previous studies have primarily focussed on the full resistance to TuMV associated with the combined mutations in both *eIFiso4G1* and *eIFiso4G2* [80,81]. Here, our analyses reveal that eIFiso4G1 is needed to support optimal TuMV accumulation. This finding is in line with previous studies which have shown that inactivating *eIFiso4G1* alone is sufficient to trigger partial resistance to the potyviruses lettuce mosaic virus (LMV) and plum pox virus (PPV) [81,82] as well as the polerovirus turnip yellows virus (TuYV) [64] in *Arabidopsis*. The reason behind this could be the higher accumulation of eIFiso4G1 than eIFiso4G2 in leaf tissues, which makes eIFiso4G1 a limiting factor. Interestingly, our data suggest an epistatic relationship between the *eIF4E1* and *eIFiso4G1* genes during TuMV infection: the absence of eIF4E1 increases the TuMV load, but the loss of eIFiso4G1 strongly reduces TuMV titers. As no functional eIFiso4G1 protein is produced in *eifiso4g1*<sup>KO</sup>, but a truncated eIFiso4G1 is present in *eif4e1*<sup>KO</sup> plants, it is tempting to suggest that the truncated eIFiso4G1 could be hijacked for TuMV accumulation. Indeed, cleaved eIF4G proteins produced on picornavirus infection are effectively recruited for viral RNA translation via internal ribosome entry sites (IRES) and confer a translational advantage for virus translation over intact eIF4G proteins [83]. Although substantially different in its length and structural complexity in comparison to picornavirus IRES, TuMV possesses a 131 nucleotide-long 5' untranslated sequence that forms secondary structures and activates reporter mRNA translation *in vitro* [84], (for review [85]). As the host factors required for the translational advantage conferred by the TuMV 5' UTR have remained unexplored, examining the potential implication of eIFiso4G1 would be an intriguing endeavour and could help fill the knowledge gap on the mechanisms of potyvirus translation [86].

Our investigation into single and combined Arabidopsis mutants for *eIF4E1* and *eIFiso4G1* has helped us address the connection between translation initiation proteins, virus accumulation, and disease symptoms in plants infected with TuMV. In particular, the simultaneous disruption of both eIF4E1 and eIFiso4G1 in the *cum1 eifiso4g1*<sup>KO</sup> mutant resulted in a striking reduction in growth and increased tissue senescence, often culminating in the collapse of the entire plant upon infection. Intriguingly, these pathological effects were not accompanied by significant viral overaccumulation, as only a limited TuMV population was detected in the living tissues of these plants. This raises a perplexing question: how can the presence of a modest amount of TuMV in *cum1 eifiso4g1*<sup>KO</sup> mutants lead to such pronounced disease symptoms? An appealing hypothesis is that TuMV could induce the degradation of the only remaining eIFiso4G2 protein in these plants, essentially leading to a complete breakdown of host functions due to the absence of canonical translation initiation complexes. Although eIFiso4G1 and eIFiso4G2 have different expression patterns, they share significant sequence similarities and functional overlap [87–90], suggesting that the mechanism responsible for eIFiso4G1 degradation might also apply to eIFiso4G2. Alternatively, and not exclusively, we could speculate

that the simultaneous inactivation of eIF4E1 and eIFiso4G1 proteins might disrupt the balance of host defense and immune response pathways. The abnormal initiation complexes formed in the absence of eIF4E1 and eIFiso4G1 could influence the translation of specific host factors that modulate disease symptom development, even if the virus titers remain low. To explore this, detailed proteomic and transcriptomic analyses are needed, comparing gene expression in plants with different combinations of inactivated initiation factors. A more focused approach could involve directly monitoring the production of proteins known to influence the severity of TuMV symptoms [91,92].

Altogether, our work here uncovers an intricate relationship between the plant eIF4F and eIFiso4F complexes that regulates host translation and potyvirus susceptibility in *Arabidopsis*. These findings provide an insight into the way potyviruses interact with the plant translation apparatus and expose weak links in strategies for the development of resistances based on translation initiation factor inactivation. Future studies are needed to understand how eIF4E1 mediates the protection of the translational machinery and whether these mechanisms can be extended to other translation initiation factors and other plant viruses.

## Materials and methods

### Plant and virus material

The *Arabidopsis thaliana* ecotype Columbia-0 (Col-0) was used as the wild-type accession and all of the *Arabidopsis* mutants used are in this genetic background. The homozygous *eifiso4g1^KO* lines were obtained by a T-DNA insertion in *eIFiso4G1* (At5g57870, SALK_009905), homozygous *eif4e1^KO* lines by a T-DNA insertion in *eIF4E1* (At1g18040; SALK_145583) and homozygous *cum1* lines by a stop codon in the *eIF4E1* coding sequence [93]. The combined homozygous *cum1 eifiso4g1^KO* mutant plants were generated through crossing followed by self-fertilization. The TuMV-GFP UK1 isolate was kindly provided by Jean-François Laliberté [94].

### Plant culture and virus inoculation

Seeds were sowed on Murashige and Skoog (MS) medium following surface sterilization for 10 min in 95% ethanol and 0.1% Tween 20. Two weeks after sowing, seedlings were individually transferred to soil and cultivated in a culture chamber at 20–24˚C with a short-day cycle of 8 hr light (100 μmol·m$^{-2}$ s$^{-1}$) and 16 hr dark.

Virus inoculation was performed on 4-week-old *Arabidopsis* plants. TuMV inoculation was done by rub-inoculating a single young leaf with TuMV-GFP UK1-systemically infected turnip (*Brassica rapa*) leaves freshly ground in phosphate buffer (0.03 M Na$_2$HPO$_4$, 0.2% diethyl-dithiocarbamate [DIECA], pH 7) containing active carbon and carborundum. Mock inoculation was done by rub-inoculating a single young leaf with the phosphate buffer alone.

### Fluorometric camera analysis

The closed fluorometric camera FluorCam FC 800-C/1010-GFP (Photon System Instruments) equipped with a GFP filter was used to monitor TuMV-GFP UK1 infection. Images were obtained by using FluorCam7 v. 1.2.5.3 software (Photon System Instruments). Fluorescence is represented in false colours from blue (low intensity) to red (high intensity).

### Polysome profiling analyses

Polysome profiling was carried out exactly as described in [51]. For absorbance profiles representation, the absorbance curves were normalized to either the monosome or to the first

polysome peak in order to compare the relative amounts of polysomes or monosomes, respectively. For absorbance profiles quantification, the area under the polysome and monosome peaks was calculated from raw absorbance profiles and expressed as a ratio (polysome area divided by monosome area) relative to mock-inoculated wild-type plants. Custom annotated R Markdown scripts describing data analyses and representations are provided in the Supporting Information.

### Puromycin labelling

Puromycin labelling was carried out as described in [55] with some modifications. Samples consisted of approximatively 200 mg of young leaves pooled from 3 to 5 independent plants of each experimental condition. The incorporation of puromycin was performed by vacuum-infiltration with labelling mixture (0.5X Murashige and Skoog salts, 0.1 g/L MES, 0.1% Tween 20, pH = 5.7) supplemented with 50 µg/mL puromycin [Apollo Scientific]. The samples were incubated with the labelling mixture in Petri dishes under growing conditions and the puromycin incorporation was stopped by freezing in liquid nitrogen after exactly 2 hours.

### m7GTP pull-down assay

The m7GTP pull-down assay was performed with minor modifications as described previously [49]. Young leaves (for western blot analysis of eIFiso4G1) or whole plants (for mass spectrometry analyses), were ground in liquid nitrogen and homogenized in a 1:3 weight to volume ratio in binding buffer (100 mM HEPES, pH 7.6, 5% glycerol, 50mM KCl, 5 mM EDTA, 0.1% Triton X-100, 5 mM DTT and 1 × protease inhibitor cocktail [Roche]). The homogenized lysates were incubated at 4°C for 1 hour for western blot analysis of eIFiso4G1 or overnight for mass spectrometry analyses with 50 µl of prewashed immobilized γ-aminophenyl-m7GTP (C10-spacer)-agarose beads (Jena Bioscience). The beads were washed one time for western blot analysis of eIFiso4G1 or three times for mass spectrometry analyses with the binding buffer at 4°C and linked proteins were eluted by boiling in 50 µl Laemmli buffer.

### Protein extraction

Proteins were extracted by homogenizing ground plant tissues in equimolar ratio of loading buffer (100 mM Tris-HCl, pH 6.8, 25mM EDTA, 4% SDS, and 20% glycerol) and heating at 85°C for 10 min. Proteins contained in the supernatant were recuperated following centrifugation for 10 min at 15,000 × g and protein concentration was determined using the BCA assay [Sigma-Aldrich]. Equal amounts of proteins were then reduced with 5% b-mercaptoethanol and were prepared for western blotting.

### Western blotting

Equal amounts of protein extracts were electrophoresed on 12% sodium dodecyl sulphate (SDS) polyacrylamide gel prior to transfer on Hybond ECL nitrocellulose membranes (GE Healthcare). Homogenous transfer was confirmed by Ponceau Red staining. Membranes were blocked overnight in Tris-buffered saline containing 5% dry milk (TBS + 5% milk) and incubated with rabbit polyclonal antibodies directed against *Arabidopsis* eIFiso4G1 (diluted 1:2000 in TBS + 5% milk) [95], mouse monoclonal antibodies (12D10 clone) against puromycin (diluted 1:1000 in TBS + 5% milk) [Sigma-Aldrich], and mouse monoclonal antibodies directed against plant actin (diluted 1:5000 in TBS + 5% milk) [Sigma-Aldrich]. Following three washes with TBS supplemented with 0.1% Tween 20, membranes were probed with secondary goat horseradish peroxidase-linked anti-rabbit serum for detection of eIFiso4G

(diluted 1:5000 in TBS + 5% milk) and a goat horseradish peroxidase-linked anti-mouse serum for detection of actin and puromycin (diluted 1:1000 and 1:5000 in TBS + 5% milk for actin and puromycin detection, respectively). Detection of horseradish peroxidase activity was performed using a LumiGLO Reserve chemiluminescent substrate kit (SeraCare) or the Substrat HRP Immobilon Western kit (Milipore) and imaged with X-OMAT LS films (Kodak) or a Fusion FX7 imager (Vilber Lourmat).

## Quantification of puromycin signal intensity

The raw 16-bit TIFF images obtained by the Fusion FX7 imager were analysed using the ImageJ analysis software (imagej.nih.gov/ij). The surface value calculated for the puromycin lane was normalized to the surface value calculated for the actin band in the corresponding sample and expressed relative to wild-type.

## Mass spectrometry-based quantitative proteomic analyses

For mass-spectrometry analyses, protein samples were prepared as described in "m7GTP pull-down assay" using γ-aminophenyl-m7GTP (C10-spacer)-conjugated agarose beads and blank agarose beads (Jena Bioscience) to obtain Cap samples and negative control samples, respectively. Three biological replicates of Cap samples using extracts from mock or TuMV-inoculated WT and $eif4e1^{KO}$ plants at 21 dpi were prepared. In parallel, two biological replicates of negative controls using the same extracts were processed. The eluted proteins were directly solubilized in Laemmli buffer, heated for 10 min at 95˚C, and stacked in the top of a 4–12% NuPAGE gel (Invitrogen). After staining with R-250 Coomassie Blue (Biorad), proteins were digested in-gel using trypsin (modified, sequencing purity, Promega), as previously described [96]. The resulting peptides were analyzed by nanoLC-MS/MS using an Ultimate 3000 RSLCnano coupled to a Q-Exactive HF (Thermo Fisher Scientific) using a 120-min gradient. For this purpose, the peptides were sampled on a precolumn (300 μm x 5 mm PepMap C18, Thermo Scientific) and separated in a 75 μm x 250 mm C18 column (Reprosil-Pur 120 C18-AQ, 1.9 μm, Dr. Maisch). The MS and MS/MS data were acquired by Xcalibur (Thermo Fisher Scientific).

Peptides and proteins were identified by Mascot (version 2.7.0.1, Matrix Science) through concomitant searches against the Uniprot database (*Arabidopsis thaliana* taxonomy, February 2020 download), the Genbank database (TuMV-K1 taxonomy November 2019 download), a homemade database containing the sequences of classical contaminant proteins found in proteomic analyses (human keratins, trypsin, etc.), and the corresponding reversed databases. Trypsin/P was chosen as the enzyme and two missed cleavages were allowed. Precursor and fragment mass error tolerances were set at respectively at 10 ppm and 25 mmu. Peptide modifications allowed during the search were: Carbamidomethyl (C, fixed), Acetyl (Protein N-term, variable) and Oxidation (M, variable). The Proline software [97] was used for the compilation, grouping, and filtering of the results (conservation of rank 1 peptides, peptide length ≥ 6 amino acids, peptide-spectrum-match score ≥ 25, allowing to reach a false discovery rate (FDR) of peptide-spectrum-match identifications < 1% as calculated on peptide-spectrum-match scores by employing the reverse database strategy, and minimum of one specific peptide per identified protein group). The mass spectrometry proteomics data have been deposited to the ProteomeXchange Consortium via the PRIDE [98] partner repository with the dataset identifier PXD036336. Proline was then used to perform the MS1 label-free quantification of the identified protein groups using specific and razor peptides.

Statistical analysis was then performed using the ProStaR software [99]. Proteins identified in the contaminant database and proteins detected in less than three replicates of one

condition were removed. After log2 transformation, abundance values were normalized using the variance stabilizing normalization (vsn) method before missing value imputation (slsa algorithm for partially observed values in the condition and DetQuantile algorithm for totally absent values in the condition). To highlight proteins enriched in Cap samples compared to negative controls, statistical testing was conducted using limma. For this, the three replicates of Cap-binding proteins were compared to the corresponding four samples of mock or TuMV-inoculated samples (i.e. wild-type mock Cap was compared to wild-type negative control + *eif4e1$^{KO}$*-negative control, wild-type TuMV Cap was compared to wild-type TuMV-negative control + *eif4e1$^{KO}$* TuMV negative control, etc.). Proteins enriched in Cap samples were sorted out using a fold change cut-off of 1 and a p-value cut-off of 0.01, leading to a FDR inferior to 5% according to the Benjamini-Hochberg estimator. Proteins found enriched in Cap condition but showing at least one missing value in this condition were manually invalidated. Proteins detected in the Cap samples in the different conditions were then compared first globally by analysis of variance (ANOVA) using Perseus [100]. The proteins were considered differentially associated with Cap through the different conditions if they show a Benjamini-Hochberg q-value ≤ 0.05. Conditions of interest were then compared two by two using limma in ProStaR. Differentially abundant proteins were sorted out using a fold change cut-off of 1 and a p-value cut-off of 0.01, leading to a FDR inferior to 5% according to the Benjamini-Hochberg estimator. Proteins found differentially abundant but detected in less than three replicates in the condition in which they were found to be more abundant were manually invalidated (p-value = 1).

### Rosette fresh weight and virus accumulation analyses

Rosette fresh weight analyses and DAS-ELISA was performed exactly as described in [49]. For quantitative DAS-ELISA, the virus load was estimated according to [101], by comparing the dilution curves obtained from serially-diluted TuMV-infected plant samples to those obtained for a common wild-type TuMV-infected plant sample loaded in each of the ELISA plates. Custom annotated R Markdown scripts describing data representation are provided in the Supporting Information.

### Data analyses

The R software (http://www.r-project.org/) was used for the majority of data analyses. Graphs were produced using the ggplot package [102]. Statistical analyses were performed using the rstatix and pgirmess packages [103,104]. Normally distributed data was analysed by Tukey's HSD test (for polysome quantification analyses) or by two-way ANOVA followed by a posthoc Tukey test (for puromycin quantification analyses). For non-parametric data (TuMV accumulation analyses by DAS-ELISA), Kruskal-Wallis tests were performed. Statistical analyses of MS-based quantitative proteomics data were performed as described in "Mass spectrometry-based quantitative proteomic analysis of m7GTP pulldown samples".

### Supporting information

**S1 Fig. Complete polysome profile absorbance data.** Comparison of monosome-normalized (left panels) and first polysome-normalized (right panels) polysome profiles for mock-inoculated wild-type and *eif4e1$^{KO}$* plants **(a)**, mock and TuMV-inoculated wild-type plants **(b)**, mock and TuMV-inoculated *eif4e1$^{KO}$* plants **(c)** and TuMV-inoculated wild-type and *eif4e1$^{KO}$* plants 14 dpi. n = 3 biological replicates. WT = wild-type, 4e1KO = *eif4e1$^{KO}$*.
(TIF)

**S2 Fig. Comparison of puromycin incorporation in mock and TuMV-inoculated wild-type and *eif4e1^{KO}* plants 14 dpi.** Western blot using anti-puromycin antibodies was performed on 20 ug of total protein extracts and equal protein levels were checked using anti-Actin antibodies. Each lane corresponds to an independent biological replicate. Membranes were imaged simultaneously.
(TIF)

**S3 Fig. Complete puromycin incorporation western blot data. (a)** Comparison of puromycin incorporation in mock-inoculated plants in 2 experimental repeats **(b)**. Quantification of puromycin signal intensity from images shown in **(a)**. Graphs show the 95% confidence intervals of the means obtained from 2 independent experimental repeats. n = 10 biological replicates. "ns" non-statistically significant difference **(c)** Comparison of puromycin incorporation in TuMV-inoculated plants in 3 experimental repeats. The photograph of the western blot associated with the third experimental repetition is displayed in Fig 1e. Western blot using anti-puromycin antibodies was performed on 20 ug of total protein extracts and equal protein levels were checked using anti-Actin antibodies.
(TIF)

**S4 Fig. Experimental repeat of Fig 3.** Western blot analyses of eIFiso4G1 accumulation in total protein extracts **(a)** or m7GTP pull down protein extracts **(b)** 14 dpi. Red arrowheads indicate the position of the full-length, 86 kDa, eIFiso4G1 protein. Dashed red arrowheads indicate the position of a 50 kDa protein product recognized by anti-eIFiso4G antibodies. Equal loading was checked on total protein extracts by western blot using anti-Actin antibodies.
(TIF)

**S5 Fig. The ~50 kDa band recognized by the anti-eIFiso4G antibodies corresponds to a truncated eIFiso4G1 protein. (a)** Coomassie stained m7GTP-pull down protein extracts separated by SDS-PAGE electrophoresis. The protein content migrating between 40 and 55 kDa was excised from the gel and subjected to MS-based proteomic analyses for peptide identification. **(b-c)** Graphic representation of a truncated eIFiso4G1 protein product migrating at ~50 kDa in m7GTP pull down extracts obtained from *eif4e1^{KO}* TuMV-inoculated plants **(b)** and mock-inoculated *eif4e1^{KO}* plants **(c)** 14 dpi. Each horizontal bar corresponds to an identified peptide spectral count matching eIFiso4G1 sequence. The positions of the eIFiso4E-binding, HEAT1/MIF4G and HEAT2/MA3 domains, as well as total peptides and spectral counts identified in each experimental condition are given. n = 2 biological replicates.
(TIF)

**S6 Fig. No strict TuMV protease consensus cleavage sequence is found on the Arabidopsis eIFiso4G1 protein sequence. (a)** A graphic representation of AteIFiso4G1 showing the positions of 10 amino acid sequences that differ by a single amino acid from the deduced consensus sequences recognized for cleavage by the Nia-Pro and P1 potyvirus proteases. For each of the amino acid sequences, the amino acid on eIFiso4G1 which differs from the consensus potyvirus cleavage sequence recognized is highlighted in red. No amino acid sequence on eIFiso4G1 bearing any similarity to the consensus cleavage sequence recognized by the potyvirus HC-Pro protease was found **(b)** A table summarizing the characteristics of the 10 amino acid sequences on eIFiso4G1 that differ by a single amino acid from the consensus potyvirus protease cleavage sequences. 'X' represents any amino acid, square brackets ([]) indicate that any of the amino acid inside is acceptable at the position, and slash (/) indicates the position of the scissile bond where the proteolytic cleavage occurs. The expected molecular weight of the potential cleavage products is shown for each of the amino acid sequences. The potyvirus

proteases cleavage consensus sites were taken from (Goh and Hahn, 2021). The amino acid cleavage consensus sites on AteIFiso4G1 (Uniprot ID Q93ZT6) were searched using Fuzzpro https://www.bioinformatics.nl/cgi-bin/emboss/fuzzpro and the expected protein sizes were calculated using the Protein Molecular Weight tool https://www.bioinformatics.org/sms/prot_mw.html.
(TIF)

**S7 Fig. Experimental repeat of Fig 4. (a)** Phenotypic comparison of TuMV-inoculated representative plants of each genotype 14 dpi. **(b)** Rosette fresh weight analysis of TuMV-inoculated plants 14 dpi. n = at least 23 biological replicates **(c)** Accumulation analysis of TuMV by DAS-ELISA 14 dpi. n = at least 23 biological replicates. Different letters depict significantly different groups identified by Kruskal–Wallis statistical tests at P < 0.05.
(TIF)

**S8 Fig. The combined absence of both eIF4E1 and eIFiso4G1 does not impair plant health in the absence of TuMV infection. (a)** Phenotypic comparison of mock-inoculated representative plants of each genotype 14 dpi. **(b, c)** Rosette fresh weight analysis of mock-inoculated plants 14 dpi in two independent experimental repeats. n = at least 8 biological replicates in **(b)** and at least 17 biological replicates in **(c)**. Different letters depict significantly different groups identified by Kruskal–Wallis statistical tests at P < 0.05.
(TIF)

**S1 Table. MS-based quantitative proteomic analysis of Cap-binding proteins in WT (Col) and *eif4e1*$^{KO}$ (KO) plants infected or not with TuMV.**
(XLSX)

**S1 Supporting Dataset. Additional supporting information contains all files related to Figs 1–4 and S1–S8 including unprocessed western blots, quantification, ELISA and weight data, MS quantification data, peptides data and scripts.**
(ZIP)

## Acknowledgments

We thank Christian Meyer, Eugenie Hébrard and Sylvie German-Retana for helpful discussions throughout this work. We are grateful to Loup Rimbaud for assistance with statistical and virus quantification analyses. DZ thanks Kyoka Kuroiwa for her invaluable advice and help on the preparation of the manuscript figures, as well as her unwavering support throughout the manuscript's production.

## Author Contributions

**Conceptualization:** Delyan Zafirov, Ben Field, Yohann Couté, Karen S. Browning, Christophe Robaglia, Jean-Luc Gallois.

**Data curation:** Delyan Zafirov, Nathalie Giovinazzo, Cécile Lecampion, Julia Novion Ducassou, Yohann Couté.

**Formal analysis:** Delyan Zafirov, Nathalie Giovinazzo, Cécile Lecampion, Ben Field, Julia Novion Ducassou, Yohann Couté.

**Funding acquisition:** Yohann Couté, Jean-Luc Gallois.

**Investigation:** Delyan Zafirov, Christophe Robaglia, Jean-Luc Gallois.

**Methodology:** Nathalie Giovinazzo, Cécile Lecampion, Julia Novion Ducassou, Karen S. Browning.

**Project administration:** Jean-Luc Gallois.

**Resources:** Cécile Lecampion, Julia Novion Ducassou, Karen S. Browning.

**Supervision:** Ben Field, Yohann Couté, Christophe Robaglia, Jean-Luc Gallois.

**Validation:** Delyan Zafirov.

**Visualization:** Delyan Zafirov.

**Writing – original draft:** Delyan Zafirov, Ben Field, Yohann Couté, Karen S. Browning, Christophe Robaglia, Jean-Luc Gallois.

**Writing – review & editing:** Delyan Zafirov, Ben Field, Yohann Couté, Christophe Robaglia, Jean-Luc Gallois.

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
