## [Decision Letter · Decision Letter 0]

25 Jul 2023

Dear Dr Gallois,

Thank you very much for submitting your manuscript "Molecular crosstalk between plant translation initiation complexes influences the outcome of virus infection" for consideration at PLOS Pathogens. As with all papers reviewed by the journal, your manuscript was reviewed by members of the editorial board and by several independent reviewers. The reviewers appreciated the attention to an important topic. Based on the reviews, we are likely to accept this manuscript for publication, providing that you modify the manuscript according to the review recommendations.

Overall, we found that your study provides important new insight into the potyvirus-host interaction at translation initiation complexes, with the indication of cleavage of eIFiso4G1 during infection in eIF4E mutants as a highlight in terms of novelty. In your revised version, we would especially like you pay attention to

- change the title so that it more efficiently draws the attention towards the concrete advances offered by your study

- include the volcano plots currently in Figure S4 in a main figure

- discuss more explicitly possible models to explain the low TuMV titers and exacerbated symptoms observed in eIF4E eIFiso4G1 mutants.

Reviewer 1 raises a number of additional points on avoidance of 4E/4G jargon and on more careful interpretation of certain results that we also ask you to please take into account while preparing your revised manuscript.

We apologize for the time it has taken to evaluate the work as it took us a while to find suitable reviewers.

Sincerely,

Peter Brodersen

Guest Editor

PLOS Pathogens

Shou-Wei Ding

Section Editor

PLOS Pathogens

Kasturi Haldar

Editor-in-Chief

PLOS Pathogens

orcid.org/0000-0001-5065-158X

Michael Malim

Editor-in-Chief

PLOS Pathogens

orcid.org/0000-0002-7699-2064

Reviewer Comments (if any, and for reference):

Reviewer's Responses to Questions

**Part I - Summary**

Reviewer #1: In spite of almost two decades of research, the interaction of potyviruses with the translation initiation machinery is far from being fully understood. In this manuscript, authors continued their work to understand the enhanced susceptibility of Arabidopsis eif4e1 mutant plants to TuMV. I found particularly interesting the apparent truncation of eIFiso4G1 in TuMV-infected eif4e1 mutant plants. While I perceive this as a rather preliminary observation that deserves deeper analyses, the whole set of data is remarkable and brings sufficient novelty.

**Part II – Major Issues: Key Experiments Required for Acceptance**

Reviewer #1: N/A

**Part III – Minor Issues: Editorial and Data Presentation Modifications**

Reviewer #1: This is an interesting and well-conceived continuation of the work reported in Zafirov et al. (2021). I found particularly stimulating the observation of the truncation of eIFiso4G1 in TuMV-infected eif4e1 mutant plants. Are there alternatives to post-translational processing to explain the appearance of the eIF4iso4G1 truncated protein? Have you looked at other stress situations different than TuMV infection, for instance, heat stress?

In general, I found the manuscript well written and well-illustrated. Here there are a number of comments and suggestions that you may be willing to consider:

Title: I found the title rather vague, please try to be more specific. In fact, your current title may apply to several of the papers already published in the subject.

Abstract:

You use 4E/4G and eIF4E/eIF4G indistinctively. I would use only the eIF4E/eIF4G notation.

Line 20: Delete called

Lines 24-25. I´m not sure that you provided sufficient evidence to conclude that “eIF4E1 plays crucial role in inhibiting the TuMV-induced degradation of the translation initiation factor eIFiso4G1”. I did not see any experiment showing an inhibitory effect of eIF4E1 over eIFiso4G1 degradation. In fact, I also doubt about the term degradation. Please, consider toning down this statement.

Results:

Lines 127-128. First subheading: The heading overinterprets the results and biases the reader. Please try to find a heading that reflects more precisely the results in this part.

Lines 148-152. I was educated in the understanding that differences that were not statistically significant were not differences. There is an effect on mutant plants, but not in WT, though a tendency can be perceived. Please re-phrase.

Line 173. Coat protein.

Line178. Fig. 1 Title: “at the expense”. Why at the expense?

Line 192. S1 Fig. Is this figure really providing information supplementary to that provided in Fig. 1? Please check the journal policy on what to include as Supplementary material.

Line 202. Panel C in S3 Fig. Is it the same as in Fig. 1e? If so, I do not see the need to include it again in S3 Fig.

Lines 215-218 and results following this experiment. Would VPg in extracts from infected plants compete with the mRNA cap in the pull-downs, masking results? Please provide arguments.

Lines 220-223. Please rephrase. If differences are not statistically significant, then there are no differences.

Lines 235-236. “the unavailability of eIF4F complexes required for protein synthesis” Add “during infection”.

Line 247. Please consider including the S4 Fig. volcano plots in the main figure, they are very informative.

Lines 264-266. But the number of counts is very low. Indeed, the ratio peptides/counts is very similar for WT and mutant (infected). Please provide convincing arguments.

Line 269. Arabidopsis.

Discussion:

Fig. 5. The diagram is nice, but I´m not sure that it is much needed; the ideas behind are rather simple, no need of a diagram to understand them.

Lines 414-415. Up to what point this response is virus-specific? Any information on what may happen during abiotic stress, for instance, heat stress?

PLOS authors have the option to publish the peer review history of their article (what does this mean?). If published, this will include your full peer review and any attached files.

Reviewer #1: No

Figure Files:

Data Requirements:

Reproducibility:

References:

---

## [Decision Letter · Decision Letter 1]

23 Oct 2023

Dear Dr Gallois,

The editors and reviewers agree that the your revised manuscript addresses the comments raised on the originally submitted version. Hence, we are pleased to inform you that your manuscript 'Arabidopsis *eIF4E1* protects the translational machinery during TuMV infection and restricts virus accumulation' has been provisionally accepted for publication in PLOS Pathogens.

Best regards,

Peter Brodersen

Guest Editor

PLOS Pathogens

Shou-Wei Ding

Section Editor

PLOS Pathogens

Kasturi Haldar

Editor-in-Chief

PLOS Pathogens

orcid.org/0000-0001-5065-158X

Michael Malim

Editor-in-Chief

PLOS Pathogens

orcid.org/0000-0002-7699-2064

Reviewer Comments (if any, and for reference):

Reviewer's Responses to Questions

**Part I - Summary**

Reviewer #1: As for the first version.

**Part II – Major Issues: Key Experiments Required for Acceptance**

Reviewer #1: The authors have addressed satisfactorily all my comments, thank you.

**Part III – Minor Issues: Editorial and Data Presentation Modifications**

Reviewer #1: The authors have addressed satisfactorily all my comments, thank you.

PLOS authors have the option to publish the peer review history of their article (what does this mean?). If published, this will include your full peer review and any attached files.

Reviewer #1: No

---

## [Editor Report · Acceptance letter]

15 Nov 2023

Dear Dr Gallois,

We are delighted to inform you that your manuscript, "Arabidopsis *eIF4E1* protects the translational machinery during TuMV infection and restricts virus accumulation," has been formally accepted for publication in PLOS Pathogens.

Best regards,

Kasturi Haldar

Editor-in-Chief

PLOS Pathogens

orcid.org/0000-0001-5065-158X

Michael Malim

Editor-in-Chief

PLOS Pathogens

orcid.org/0000-0002-7699-2064